# The Economic Cost of Tobacco Farming in Bangladesh

**DOI:** 10.3390/ijerph17249447

**Published:** 2020-12-17

**Authors:** AKM Ghulam Hussain, Abu Shara Shamsur Rouf, Shafiun Nahin Shimul, Nigar Nargis, Tara Mona Kessaram, Syed Mahfuzul Huq, Jagdish Kaur, Md Khairul Alam Shiekh, Jeffrey Drope

**Affiliations:** 1Department of Economics, University of Dhaka, Dhaka 1000, Bangladesh; 2Department of Pharmaceutical Technology, University of Dhaka, Dhaka 1000, Bangladesh; rouf321@yahoo.com; 3Institute of Health Economics, University of Dhaka, Dhaka 1000, Bangladesh; shafiun.ihe@gmail.com; 4American Cancer Society, Atlanta, GA 30303-1002, USA; nigar.nargis@cancer.org; 5World Health Organization, Dhaka 1000, Bangladesh; kessaramt@who.int (T.M.K.); huqs@who.int (S.M.H.); 6WHO South East Asia Regional Office, Delhi 110001, India; kaurj@who.int; 7National Tobacco Control Cell, Ministry of Health & Family Welfare, Government of Bangladesh, Dhaka 1000, Bangladesh; khairulssz@gmail.com; 8School of Public Health, University of Illinois at Chicago, Chicago, IL 60607, USA; jdrope@uic.edu

**Keywords:** economic cost of tobacco farming, environmental cost of tobacco farming, alternative livelihoods

## Abstract

The extent of tobacco cultivation remains substantially high in Bangladesh, which is the 12th largest tobacco producer in the world. Using data from a household survey of current, former, and never tobacco farmers, based on a multi-stage stratified sampling design with a mix of purposive and random sampling of households, this study estimated the financial and economic profitability per acre of land used for tobacco cultivation. The environmental effects of tobacco cultivation on land and water resources were estimated using laboratory tests of sample water and soil collected from tobacco-cultivating and non-tobacco cultivating areas. The study finds that tobacco cultivation turns into a losing concern when the opportunity costs of unpaid family labour and other owned resources, and the health effects of tobacco cultivation are included. Tobacco cultivation poses a significantly high environmental cost that causes a net loss to society. Nevertheless, the availability of unpaid family labour and the options of advanced credit as well as a buy back guarantee from the tobacco companies attract farmers to engage in and continue tobacco cultivation. Therefore, supply side interventions to curb the tobacco epidemic in Bangladesh need to address major drivers of tobacco cultivation to correct the wrong incentives and motivate tobacco farmers to switch to alternative livelihood options.

## 1. Introduction

Tobacco use is one of the major risk factors for noncommunicable diseases. About 46.0% of adult males and 25.2% of adult females in Bangladesh use tobacco [1]. In 2018, Bangladesh experienced nearly 126,000 deaths caused by tobacco-attributable diseases, accounting for around 13.5% of all-cause deaths in the population [2]. Approximately 1.5 million adults were suffering from diseases attributable to tobacco use and nearly 61,000 children were suffering from diseases due to exposure to second-hand smoke. Tobacco-induced deaths and diseases alone cost the economy of Bangladesh around BDT 305.6 billion (USD 3.61 billion) a year, which was equivalent to 1.4% of its national GDP in the year 2017–2018. 

To meet the unyielding demand for tobacco leaf from both domestic and foreign manufacturers of tobacco products, the extent of tobacco cultivation remains considerably high. Bangladesh is the 12th largest tobacco producing country in the world [3,4,5]. Tobacco farming is growing fast and competing for the limited and fixed arable land of 37,674,000 acres. While in 2007–2008, a total of 72,000 acres of land was used for tobacco cultivation, it increased to 127,000 acres by 2014–2015—a 74% increase over seven years [6]. 

The tobacco industry publicly makes the claim that tobacco cultivation generates gainful employment and income for tobacco farmers, while global evidence suggests otherwise. Tobacco cultivation is highly labour intensive, as evident in similar studies conducted for Kenya [7], Indonesia [8], and Zambia [9]. During curing, tobacco farmers have to work 70 h at a stretch [10]. The use of unpaid family labour is widespread. A significant number of women and child labourers are engaged in tobacco cultivation and tobacco leaf processing, which constitutes exploitation of women and children and is against the law [11]. Child labour should be prohibited under the International Convention for the Elimination of the Worst Forms of Child Labour. In addition, child labour leads to loss of human capital because children cannot go to school while they are at work. Working in tobacco growing also poses serious health hazard to children. Tobacco farmers are at high risk of becoming sick with diseases, such as bronchial asthma, lung cancer, acute breathing problem, nausea, dizziness, acute nicotine poisoning popularly called green tobacco sickness (GTS) [1], respiratory disorder, musculoskeletal injuries, mental disorders and other health issues [12,13].

Along with generating health hazards for tobacco growers, tobacco cultivation leaves significant detrimental effect on the surrounding environment, including deforestation resulting from the demand for wood to cure tobacco leaves [14], degradation of soil fertility, pollution of ground water, pollution of surface water, and adverse impacts on surrounding trees and animals, that is, the biodiversity of the country, due to the use of massive amounts of chemical fertilizer and pesticides in tobacco cultivation. Tobacco plants sap soil nutrients at a faster pace than most other crops. For example, tobacco consumes two and a half times more nitrogen, seven times more phosphorus and eight times more potassium compared to maize [12]. Tobacco cultivation not only leads to the shrinkage of cultivable land for food grains, but also causes extreme environmental pollution along with health hazards and diseases. 

Even with so many negative effects, tobacco cultivation is keeping its pace in Bangladesh, partly because, like in many other countries, tobacco farmers in Bangladesh are provided with soft loans, inputs, and forward prices by the tobacco industry for their produce. Poor farmers, constrained by cash and both fixed and working capital, are easily lured by these incentives for their survival. Furthermore, tobacco is considered a normal agricultural good and a cash crop and therefore it enjoys almost all sorts of government subsidies such as lower electricity tariff for irrigation and support meant for poor and marginal farmers. As a result, the private cost of tobacco farming in many instances remains understated. If we account for all the social costs (e.g., subsidies, cost of deforestation and environmental degradation), the total gains over the competing crop production will wither away. In a country with limited land resources and food deficit, the alternative use of land for anything other than essential and staple crops tends to threaten the long-term food security of the nation.

Tobacco farmers are encouraged by the tobacco industry to continue and expand tobacco cultivation by offering various incentives including loan and buy back guarantee. On one hand, farmers become entrapped in a vicious cycle of debt, and on the other hand, they consider tobacco farming profitable without taking into consideration the private costs, including unpaid family labour, health hazards caused by tobacco handling as well as the social costs, including environmental damage to soil, water, and forest resources imposed by tobacco cultivation.

Existing scientific studies have, however, not considered the harmful effects of tobacco cultivation on the environment. The exclusion of environmental cost from the calculation of the costs of tobacco cultivation has resulted in incomplete assessment of the true cost to society. The present study fills this gap by measuring the environmental costs of tobacco cultivation, as well as the implicit costs (e.g., unpaid family labour, health costs), alongside the accounting costs of cultivation in a quasi- experimental set up.

Tobacco farming is more popular among farmers in the lowest socio-economic status [15]. Measuring the burden of tobacco farming on society in general, and the poor in particular, is therefore a priority in moving the development agenda forward, especially in view of Sustainable Development Goals 1 (End poverty in all its forms everywhere), 3 (Ensure healthy lives and promote well-being for all at all ages), 6 (Clean water and sanitation), 8 (Decent work and economic growth), 12 (Responsible consumption and production), 13 (Combat climate change and its impacts), 14 (Life below water: Conserve and sustainably use the oceans, seas and marine resources), and 15 (Protect, restore and promote sustainable use of terrestrial ecosystems, combat desertification, and halt and reverse land desecration and halt biodiversity loss) [16].

Drope et al. [8] conducted a study on the economic impact of tobacco cultivation for Indonesia. Although there is a myth about higher return in tobacco cultivation, the study results do not support that notion. The study found that tobacco cultivation is not profitable for most of the farmers, and the farmers often misperceive or miscalculate the actual profits. In addition, farmers ignore the larger opportunity costs of family labour and other similar costs. It is also evident that, even with harder work, the farmers’ fate does not change much as they remain poor and many of them have to rely on government assistance. Sadly, child labour is more prevalent in tobacco cultivation compared to the cultivation of other crops. A study also finds that tobacco farming is negatively related to income ceteris paribus [15]. All these are manifestations of a grim picture of tobacco cultivation for farmers’ livelihoods. 

Similar types of studies were conducted in Zambia [9] and Kenya [7]. The Zambian study echoes the findings of Indonesian studies such as the evidence of the tobacco farming being less lucrative and farmers remaining poor even with several years of tobacco cultivation. Moreover, the farmers get stuck in tobacco farming due to the contracts with the tobacco companies which push them into perpetual debt. Paradoxically, most of the farmers mention availability of credit is one of the reasons why they continue tobacco cultivation. Unlike the Indonesian study, this study pays considerable attention to contract farming, a very common feature in the tobacco farming sector. The farmers earn a positive profit if they only consider input costs ignoring labour, especially unpaid family labour costs. The inclusion of family labour costs makes this venture unprofitable, and this is even more unprofitable under contract settings, since contract farming requires more labour hours which include a great amount of family labour. Kenyan study also finds similar results—tobacco cultivation is profitable only if unpaid family labour costs are excluded. Since contract farming requires more extensive and intensive work, the opportunity cost is even higher, making the tobacco cultivation an unprofitable undertaking. 

All these three studies mentioned are very extensive in terms of exploration of tobacco farmers’ livelihood and their characteristics. However, one major limitation of these studies is that they compared the economic costs of current and former tobacco farmers in the tobacco-growing areas without consideration of never tobacco farmers from either tobacco-growing or non-tobacco growing areas as a control group. It is possible that the regions from where data were collected might be responsible for low income from tobacco cultivation. Unless we know what the counterfactual would have been with cultivation of other non-tobacco crops by never tobacco farmers, the estimates of the economic costs of tobacco farming might be biased. The present study seeks to correct this bias. The study on Indonesia overcomes this limitation to some extent by comparing the economic costs between current and former tobacco farmers. Moreover, even though the studies mention the detrimental effects on health and environment, limited effort has been made in estimating those effects. Therefore, these studies are confined to the direct and indirect private costs and benefits only. 

One previous study conducted by Akter [10] was very detailed in identifying various reasons and profitability in tobacco cultivation. However, this study lacked any estimate of the environmental impacts. It was primarily a qualitative study and therefore less representative. Ali et al. [12] attempted to understand the health and environmental effects of tobacco cultivation in Bangladesh. They found that tobacco cultivation leaves negative effect on the health of the farmers, and there is also environmental effect in the form of air pollution. They did not, however, estimate the economic costs of those effects. Moreover, the study was of very small scale. A more recent study by Karim et al. [17] attempted to understand the health and environmental impacts while Kutub & Falgunee [18] discussed about environmental degradation. All of these studies are mostly exploratory and are of very small scale. While all of the studies discussed about the detrimental effects of tobacco cultivation on the environment and regarded it as an important dimension, no attempt has been made to measure or estimate it. Without these estimates, the social benefit cost analysis which is critically important for decision making by policymakers is not possible. By estimating the environmental costs, this study fills the evidence gap on the societal costs of tobacco cultivation. 

To support policy makers to adopt appropriate tobacco control measures, reliable information on the economic costs of tobacco including environmental costs of tobacco farming is crucial. The WHO Framework Convention on Tobacco Control (FCTC), through article 18, calls for the protection of environment and health of persons involved in tobacco cultivation, and through article 17, provides emphasis on alternative livelihoods to tobacco farmers and workers [19]. Bangladesh ratified the WHO FCTC in 2004 [19] and is therefore committed to take steps in these regards. The detrimental effects of tobacco use on human health and the economic cost of tobacco-related illnesses in Bangladesh is well documented [2]. However, there is lack of robust scientific research on the measurement of the economic cost of tobacco farming including its environmental and opportunity costs, except for some small scale and less rigorous studies [17,18,20]. Moreover, most of the previous studies carried out in Bangladesh have failed to take into account all dimensions of environmental impacts as well as the implicit costs (e.g., unpaid family labour) making the results less convincing, incomplete or underestimated. The estimates of the economic cost of tobacco farming obtained from this study will inform policymakers about the extent of the fast-growing problem of tobacco farming in the country and provide a scientific basis for pro-active tobacco control including curbing tobacco cultivation.

## 2. Methodology

This is a cross-sectional study that included tobacco farmer households (treatment group) classified by tobacco farming status, such as current and former tobacco farmers and never tobacco farmers (control group), from both tobacco and non-tobacco cultivating areas. The study was undertaken by the Bureau of Economic Research of the University of Dhaka; which received funding and technical support from the World Health Organization (WHO) and technical support from the American Cancer Society (ACS). 

### 2.1. Survey Design and Data Collection

The primary data were collected from a multi-stage stratified sampling design with a mix of purposive and random sampling of households of current, former and never tobacco farmers using a structured questionnaire through face-to-face interviews by trained interviewers. The samples of current and former tobacco farmers were selected from two purposively selected major tobacco growing Upazilas (lowest level of administrative unit) in each of four purposively selected major tobacco-growing districts (Bandarban, Kustia, Lalmonirhat, and Manikgonj). The sample of never tobacco farmers was selected from both the above tobacco farming Upazilas and four non-tobacco farming Upazilas randomly selected from four randomly selected non-tobacco farming districts (Barisal, Sylhet, Rajchahi, Mymensingh). The tobacco-growing areas, level of production, and study sample areas are shown in the maps of Bangladesh in Appendix A
Figure A1, Figure A2 and Figure A3.

Inclusion of both tobacco and non-tobacco growing areas gave us the opportunity of examining the differences in costs between the treatment and the control groups situated within tobacco growing areas and beyond it. It allowed us to isolate the external effects of being in the neighbourhood of tobacco growing areas for never tobacco farmers. For example, the contamination of soil and water in the tobacco growing areas can raise the cost of production and reduce the return of growing other crops in the tobacco growing areas for farmers regardless of their tobacco farming status.

Under the overall supervision of the principal investigator and the co-investigators, and the direct supervision of the field coordinator, interviewers collected data from households, with support from the local health authorities. The head of the family or any other adult family member having enough knowledge on tobacco/paddy farming and marketing by his/her family was interviewed from each household. Only paddy growing was taken into consideration as the principal economic activity of the control group of never tobacco farmers. The interviewers visited each selected household three times to complete the interview before declaring it ‘not completed’.

As the objective of the study was to estimate the cost and return to tobacco cultivation, we used available data on profit made by tobacco farmers to calculate the sample size of farmer households that would be sufficient to estimate the population mean of annual profit with precision. On average, a tobacco farmer makes a profit of BDT 60,000 annually from harvesting one hectare of land [21]. The sample size was obtained using the formula [22]: $n=\frac{z^{2}\sigma^{2}}{d^{2}}$, where *n* = required sample size; *z* = 1.96 (critical value for 95% confidence level); *σ* =population standard deviation = BDT 15,000; *d* = margin of error (2.5% of mean profit) = BDT 1500. Therefore, *n* = (1.96 × 1.96) × (15,000 × 15,000)/(1500 × 1500) = 384. Adding 20% nonresponse, *n* = (384/0.8) = 480. Therefore, the study targeted interviewing 480 current tobacco farmers from 480 tobacco farming households. Similarly, 480 former tobacco farmers and 480 never tobacco farmers were planned for interview from the tobacco-growing areas.

When we consider tobacco farmers, former tobacco farmers and never tobacco farmers as mutually exclusive groups, then 480 households from each group might appear to be a reasonable sample size. However, upon piloting, we realized that former and never tobacco farmers possessed similar characteristics, and so collecting a substantial number of samples from former tobacco and never tobacco farmers will not add much value in terms of precision. Rather, having more samples from the current tobacco farmers appeared to be more useful. Therefore, we decided to increase the sample size for the current tobacco farmer group keeping total samples of all groups very much like the initial plan. In addition, 240 samples of never tobacco farmers from the non-tobacco-growing areas were selected for interview. The final total sample size, after allowing for data validation and cleaning, was 1549 comprising 625 current, 327 former and 597 never tobacco farmers. The detailed distribution of final sample size by districts is provided below (Table 1).

The survey collected two sets of data. First, household level data were collected on demographics, socio-economic profile of respondents, data related to farming including resource endowments, land allocation, labour allocation, other technological and management decisions, costs related to farming including labour costs, other factor prices, environmental factors (physical), asset distribution of the farmers, area of tobacco and paddy farming land, condition of land, loan facility, input requirement, trend in input use, tobacco processing related data, such as wood, labour, price of raw tobacco, finished tobacco, and profitability of tobacco and paddy cultivation.

Second, to understand the impact of tobacco cultivation on soil and water quality, soil and water samples were collected from both tobacco-cultivating (treatment) and non-tobacco-cultivating (control) areas by trained collectors of specimen suitable for laboratory analysis. The soil and water samples were tested in scientific laboratories of the Department of Soil Science and Pharmaceutical Technology of the University of Dhaka to evaluate the impact of tobacco cultivation on the chemical composition of soil and water in tobacco cultivating areas.

The primary data were supplemented with secondary data collected from documents published by the Government agencies, including Bangladesh Bureau of Statistics (BBS) and Ministry of Agriculture, and various studies related to the environmental effects of cultivation on air, water, and soil quality.

### 2.2. Analytical Framework

The study estimated the profitability of tobacco cultivation per acre of land in three stages. First, the financial profit per acre was estimated by using the difference of the sales revenue of tobacco leaf and the costs of explicit inputs used in tobacco cultivation. Second, the implicit costs of tobacco cultivation, such as the opportunity costs of unpaid family labour, own land and fuelwood from own sources, were estimated and subtracted from the estimated financial profit to obtain the estimate of private economic profit per acre. Finally, other indirect costs, including healthcare costs, costs of environmental impact through carbon emission from wood burning, deforestation, and sequestration, and detrimental effects of pesticide and chemical fertilizer on soil and water bodies in tobacco cultivating areas were estimated and subtracted from the private economic profit to obtain the estimate of social profit per acre.

## 3. Measures

### 3.1. Cost of Deforestation Attributable to Tobacco Farming

We estimated the total wood required to process different types of tobacco in Bangladesh. The total wood required was then converted to equivalent measure of deforestation, and the magnitude of deforestation was converted into dollar value. The calculation of solid wood required for tobacco processing is based on the following formula: C = F × H × M, where C is the quantity of firewood used in tons per year, F is the quantity of firewood in tons required to cure one metric ton of tobacco leaves, H is the average production of tobacco leaves in tons per hectare of tobacco cultivated land, M is the total number of hectares of tobacco cultivated land [23]. Then, C is converted to cubic meters of firewood using a standard coefficient [24]. The measures of F and H were obtained from the survey, and the data on M was based on a secondary source [23].

A stack of wood is one metre long by one metre wide by one metre high, giving a total volume of one (stacked) cubic metre (stm^3^). Due to irregular gaps and air spaces, only approximately 60–70% of the volume is made up of solid wood, so that the weight of wood in one stack will range from approximately 250–600 kg. This translates into a mean stacking factor of 425 (kg) or 0.43 (tonnes). An equivalent ratio of 2.33 t could thus be used to convert solid wood into a stack of wood [25].

The next step is to translate the annual amount of solid wood required (in tonnes and in cubic meter) into the equivalent area of woody biomass needed (in hectares) and either managed on a sustainable basis using the mean annual increment (MAI) approach or deforested using the growing stock (GS) approach. MAI means the annual increase in the aggregate volume of trees, commonly expressed in solid volume per hectare. It is often used to indicate the yield, since it represents the long-term sustainable quantity of wood which can be harvested. MAI specifications have fixed range and are dependent on the temperate plantation and tropical plantation. The tropical plantations normally give an MAI in the range of 6–24 m^3^ /ha per year (mean = 15 m^3^ /ha) [26]. Using the MAI values as specified, the equivalent area of woody biomass needed and assumed to be harvested on a sustained-yield basis was calculated.

The growing stock (GS) of woody biomass is the (solid) volume of wood standing on a given area such as one hectare (0.01 square km). The estimated coefficient of woody mass per hectare of land has a range. For example, pine forest has 85–120 M^3^ of pine wood per hectare. Deforestation means complete removal of the natural woody biomass (depletion of growing stock). GS specifications are fixed and dependent on the ecosystem of a continent. Using GS values as specified for forest and woodlands, the wooded area needed to be completely removed (deforested) was calculated.

Once the total equivalent deforestation area in hectares was obtained, the carbon sequestration approach was used to estimate the environment related costs. The carbon sequestration method estimates the emission of carbon to the environment from burning the wood, and dissipation of carbon from soil due to deforestation. Firstly, it is standard that about 50% of biomass is the amount of carbon emitted from burning wood. From the data already available on total amount of wood burnt annually in metric tonnes, the amount of carbon emitted was calculated.

Secondly, sequestration is the stock of organic carbon stored in the soil. Carbon is emitted naturally from the soil and it is accelerated due to deforestation. It is found that the standard rates of emission of carbon as CO_2_ from soil are 975 t/hectare from bare-land and 868 t/hectare from arable land in 20 years. Using the total equivalent area of deforestation obtained in the previous step, we calculated the total carbon as CO_2_ emitted from soil due to deforestation caused by tobacco cultivation.

Adding the amount of carbon emitted due to burning of firewood and the amount of carbon emitted due to deforestation, the total amount of carbon emitted was calculated. There is substantial variation in the value of environmental costs of emission ranging from USD 100 to 1000 [26]. However, we used a very conservative measure as suggested by the EPA in 2019 [27] to translate the amount of carbon emitted to dollar figure.

In Bangladesh, tobacco cultivation mainly replaces other essential crops like paddy. The deforestation, therefore, mainly takes place to cure and process tobacco leaves, and therefore land cleared of forest to grow tobacco was not considered.

### 3.2. Cost of Soil and Water Quality Degradation Attributable to Tobacco Farming

There is no established study or technique for estimating the cost of topsoil contamination and water degradation attributable to tobacco cultivation. This study is the first of its kind to quantify the cost of damage of water quality and soil quality by tobacco farming. We measured the degree of contamination due to use of pesticides and fertilizers as well as the depletion of nutrients due to tobacco cultivation. Based on information from secondary sources, we assessed the total cost to restore the soil and water condition in tobacco cultivating areas to the level in the control areas (paddy cultivating area). For that purpose, 80 soil samples, 80 surface water samples, and 32 ground water samples (from tube well) were collected from tobacco growing fields and nearby water bodies of 16 tobacco growing Upazilas. Further, 72 soil samples, 72 surface water samples and 24 ground water samples (from tube well) were collected from paddy growing fields and nearby waterbodies of 24 non-tobacco growing Upazilas.

Soil samples were collected from 0–15 centimetres (topsoil) in clean polythene bags, properly labelled, and taken to the labouratory for analysis. Topsoil has the highest concentration of organic matter and microorganisms and is where most of the biological soil activity occurs. The quality of soil from tobacco cultivation and control areas, in terms of mineral particles, organic matters, and pesticides, was compared. The variables used to assess the soil quality include pH Level, phosphate (PO4), total nitrogen (N%)/nitrate (NO3), exchangeable potassium (K^+^), organic carbon (OC%), magnesium (Mg), nickel (Ni), zinc (Zn), copper (Cu), aldicard, chlorpyrifos, and 1,3-D (1,3-dichloropropene, also known as Telone).

Water samples were collected from nearby ponds, canals (surface water) and tube wells (ground water) of tobacco farming and control areas to check its physical, chemical and biological quality.

The variables used to assess the water quality included water clarity (turbidity and total suspended solids), dissolved oxygen (to be measured by DO meter), water temperature, pH, specific conductance (the ability of water to pass an electrical current), nitrates (NO3), phosphorus (P), potassium (K), aldicard, chlorpyrifos, and 1,3-D [28].

To come up with an estimate of the damage to soil and water from tobacco cultivation, we used secondary sources to get the parameters to restore the soil and water quality to the level observed in the control areas. We also estimated the loss of fisheries and faunas due to water contamination from tobacco cultivation compared to the control areas, and the loss of grazing land by comparing livestock density in tobacco growing and control areas. Primary data collected from household survey and secondary data (on fishery and livestock) were used to obtain the estimates.

Adding the soil and water quality reversal cost, the fish stock loss due to water quality degradation and the economic loss from anticipated lower production of livestock in the tobacco cultivating area, we estimated the environmental cost due to water and soil quality degradation.

### 3.3. Household Level Profitability of Tobacco Farming

The financial profit made by the farmers was estimated by using the difference of the sales revenue of tobacco leaf and the costs of explicit inputs used in tobacco cultivation, including fertilizer, seed, and hired factor (labour) costs. The private economic profit from tobacco farming was estimated by the difference in financial profit and implicit costs, such as the opportunity costs of unpaid family labour, own land, and fuelwood from own sources. In case of family labour, market wage rate in tobacco farming was used to estimate the opportunity costs, and market rate of leasing cost for land was used to measure forgone opportunity costs of own land use. Similar approach was applied for estimating the implicit costs of fuelwood used from own source. The productivity lost due to sick days was used to estimate the opportunity cost of time lost due to sickness. The accounting of health care spending for treatment of illnesses was, however, beyond the scope of this study.

### 3.4. Social Profitability of Tobacco Farming

The tobacco farming-attributable average environmental cost (TFAEC) per acre was estimated using the following model:TFAEC = CECFWB + CSCFDF + CSQR + CWQR + LFF(1)
where

CECFWB = average cost of carbon dioxide emission from wood burningCSCCFDF = average cost of carbon sequestration from deforestationCSQR = average cost of soil quality reversalCWQR = average cost of water quality reversalLFF = average cost of loss of fish and fauna

Then, combining all costs, both direct (i.e., material input cost and hired labour) and indirect (i.e., cost of deforestation, degradation of soil and water quality and loss of fish and fauna due to tobacco farming, health costs, costs of family labour), we finally estimated the social cost of tobacco farming per acre of land. Subtracting this cost from the revenue earned by tobacco farming per acre resulted in social profit or the net economic gain of tobacco farming per acre.

### 3.5. Comparison of Private Economic Profit of Tobacco with Paddy

The private economic profit from tobacco farming was compared with that of paddy cultivation, considering that paddy would have been farmed had tobacco been not grown in the available land. This provided an opportunity to estimate the opportunity cost of tobacco farming over paddy cultivation or the gain from shifting from tobacco cultivation to other major crops.

### 3.6. Cashpor Housing Index

The study used the Cashpor housing index (CHI) to determine the socio-economic status (SES) of the respondents. CHI is believed to be one of the cost-effective ways for identifying the poor. It uses housing condition to understand the socio-economic status of the respondents. Since low income households spend substantial amounts of household resources on daily necessities and other social obligations (such as weddings of children), they do not have enough saving to invest in improving housing conditions. Though a good housing condition helps to increase social status, not being able to invest in housing indicates that the residents are poor. This index is widely used in the context of developing countries (such as in [29,30]).

## 4. Findings

### 4.1. Characteristics of Sample Households

Current tobacco farmers (contract and independent), former tobacco farmers, and never tobacco farmers (both from tobacco farming and non-tobacco farming districts) possess very similar characteristics in terms of maximum level of education of household members, household size and primary occupation (Table 2). The average maximum level of education of the farmer households is just above nine years; average household size is close to five including all members and less than four including members above age 12 only. Though the majority of both tobacco and non-tobacco farmers have farming as the primary occupation, a somewhat higher percentage of current tobacco farmer households (79.24%) reported farming as their primary occupation than former tobacco farmers (72.31%) and never tobacco farmers (75.95%). Agriculture being the primary occupation among a higher proportion of the current tobacco farmers might indicate limited diversity in income sources which restricts their choice to tobacco cultivation to ensure certain and consistent flow of income.

Based on the Cashpor index, 32% of the current tobacco farmers fall under the high SES category, while 35% of never tobacco farmers belong to this group (Table 3). However, 39% of the never tobacco farmers belong to the lowest SES category, while 33% of current tobacco farmers belong to this group. It suggests that tobacco farming might have helped tobacco farmers graduate to the middle SES from the lowest SES, but this force is not strong enough to push farmers to be in the highest SES. This phenomenon is further substantiated by the smaller percentage of high SES category among the former tobacco farmers. Even though it is often believed that tobacco farming brings long-term prosperity, the lower share of former tobacco farmers in the high SES group does not attest to this view.

The average size of total landholding of the current tobacco farmers (145.96 decimals, i.e., 1.46 acre) is lower than the never tobacco farmers (154.80 decimals., 1.55 acre), but higher than the former tobacco farmers (107.95 decimals) (Table 4). Out of 145.96 decimals, tobacco farmers cultivate tobacco in 117.78 decimals (81%) of land.

While the proportion of cultivation in own land is similar (50% vs. 52%) between the contract and independent tobacco farmers, some noticeable variations exist in the proportion of land under sharecropping (44% vs. 20%) and cultivation with lease (5.57% vs. 28%) between these two groups (Figure 1). The share of own land is larger among never and former tobacco farmers than current tobacco farmers, which may indicate that tobacco farmers prefer to lease land or have recourse to share cropping for tobacco cultivation considering the adverse effect of tobacco cultivation on the soil quality and productivity of land.

Current tobacco farmers have lower household monthly income and higher household monthly expenses and assets compared to the never tobacco farmers (Table 5). The average household income of independent tobacco farmers is higher than the contract tobacco farmers, which may seem counterintuitive. As the household income variable comprises income from all sources including remittance and non-farming activities, this dichotomy indicates that contract tobacco farmers may have less alternative income opportunities. For some farmers, tobacco is the only crop they know how to grow, which may be a good reason for them to assent to contractual arrangements with tobacco companies. This study shows that contract farmers, on average, owe more than BDT 19,000 in loans from tobacco companies.

Even though the number of work hours per day, number of days per month and the number of months per year in non-farm employment do not vary significantly across current, former and never tobacco farmers, the total annual work hours is significantly higher for former and never tobacco farmers in non-tobacco growing areas compared to current tobacco farmers. The high labour intensity of tobacco farming may limit the hours that tobacco farming households can allocate to other non-farm income generating activities. As shown in the Table 6, the average monthly pay for former tobacco farmers from non-farm employment is higher than current tobacco farmers, indicating the presence of an incentive to move to non-farm employment away from tobacco growing.

### 4.2. Input Costs

The cost of inputs, namely fertilizers, pesticides, and other raw materials, is significantly higher for tobacco farmers than the costs of inputs for non-tobacco farmers (Table 7). Moreover, inputs used in absolute terms are also much higher for the tobacco farmers. There is also a significant variation in the average input cost between contract and independent tobacco farmers, as contract farmers spend a lot more on fertilizer, pesticides, and other variable inputs compared to independent farmers. Less variation is observed between the cost of durables for independent tobacco farmers and never-tobacco farmers. However, contract farmers also spend enormous amount on durables. They spend almost three times more in fertilizer than the never tobacco farmers. It is common knowledge that tobacco farming requires more fertilizer and pesticides, but contractual arrangements seem to impose a greater use of these inputs. Continuous monitoring and target setting by the tobacco companies might force the tobacco farmers to act in such a way.

In tobacco cultivation, especially during curing leaves, farmers must work for weeks at a stretch, and so the use of family labour is very common. This study finds substantial use of family labour for tobacco cultivation. While yearly family labour hours used by never tobacco farmers is 934, it is 722 h more for tobacco cultivation, and for contract tobacco farmers the difference is even higher (Table 8). The use of hired labour is also higher for tobacco farmers resulting in higher labour costs. Among current tobacco farmers, contract farmers use both family and hired labour hours more than independent tobacco farmers. Moreover, the wage rate in contract tobacco farming is the highest among all categories, which might be a sign of both extensive and intensive use of family labour in tobacco farming especially under contract farming. Since an enormous amount of family labour hours remain unpaid, the monetary labour costs of tobacco cultivation underestimate the actual labour costs. Though use of unpaid family labour hours is common for non-tobacco farmers as well, labour cost in tobacco cultivation is substantially higher than that in the cultivation of non-tobacco crops considering the opportunity cost of family labour.

### 4.3. Private Financial Profitability of Tobacco Farming

Table 9 reports the average cost of production (including input cost, hired labour cost and land rent cost) and sales revenue of crops and the average profit and rate of profit (profit as percentage of cost) per acre of land calculated by the authors based on the cost and revenue data. It shows that tobacco cultivation costs tobacco farmers more than other crops cost former and never tobacco farmers. Although the sales revenue is higher for tobacco farmers than former and never tobacco farmers, the financial profit is lower due to higher input costs for the former than the latter groups (BDT 16,463 for current tobacco farmers vs. 44,444 for former tobacco farmers and 34,239 for never tobacco farmers). The profitability is much lower specifically for independent tobacco farmers. The rate of profit as a percentage of the cost of production is several times larger for former and never tobacco farmers than for current tobacco farmers, indicating that the rate of return on investment in non-tobacco crop cultivation is much larger than in tobacco cultivation, providing sufficient evidence to incentivize tobacco farmers to move to non-tobacco alternative crop cultivation. Hence, the myth of higher return from investment in tobacco farming is not supported by the data.

In the last few years, the price of paddy especially during harvest times was low. Many marginal farmers usually sell the paddy during that season and sales revenue could be highly sensitive to prices. The negative profit for never tobacco farmers in non-tobacco growing areas is likely subject to this seasonal fluctuation at the time of the survey. In addition, we do not have counter-factual data for that area so there is no way to know what the profit would have been if tobacco were cultivated in that area. Therefore, a comparison of non-tobacco farmers of tobacco areas with tobacco farmers of those same areas appears to be more reasonable.

### 4.4. Health Costs

Tobacco cultivation not only requires extensive labour of household members, it also exposes them to health hazards from handling tobacco. Table 10 shows that the average number of sick days of household members was higher for current tobacco farmers than never tobacco farmers. The former tobacco farming households experienced even larger episodes of sickness indicating the long run detrimental effects of tobacco cultivation on health. As a result, health costs, both direct (e.g., doctor’s fee, medicine, hospital stay, transportation, etc.) and indirect (e.g., loss of working days and income), are higher for both current (11% higher) and former tobacco farmers (50% higher) in comparison with the never tobacco farmers (Table 10).

### 4.5. Environmental Costs

Various forms of contamination were observed in water and the level of contamination was higher for water collected from the water bodies adjacent to tobacco cultivating land. Various forms of soil contamination were also evident in tobacco cultivating land. The most important parameter significantly higher in tobacco cultivating land was the presence of toxic aldicarb. A neutralization process of aldicarb is proposed with a cost of USD 0.0002 per kg of soil. We estimated the cost of treatment of soil to revert the soil quality to the aldicarb level of other crops. The cost per bigha (approximately one-third acre) was estimated at BDT 5498 (weight of soil in one-cubic feet is 100 pounds (https://www.reference.com/science/much-cubic-foot-soil-weigh-7b509c02c5101291). One bigha is equivalent to 14,400 square feet or one-third acre). Since 6″ depth of soil is considered for treatment, the total soil amount per bigha to be treated is 0.5 × 14,400 cft. The cost of treatment is 0.0002 dollar per kg of soil. The cost per bigha is equal to (0.0002 × 0.5 × 14,400 × 100 × 84)/2.2 = BDT 5498 (the cost per acre is equal to BDT 15494).

Additional environmental cost originates from curing of tobacco leaves. About 50% of the tobacco leaves are dried/cured with direct sun, and 48% of them are cured through fuel (Table 11). Wood is the main fuel used for curing followed by bamboo. About 94% of leaves that are fuel-cured are cured with wood which is primarily purchased from the market. To cure 40 kg (famers in Bangladesh are used to the local measure of mound which is equivalent to 40 kg) of leaves, 201.77 kg of wood is required with an average price of 40 kg woods at BDT 147.45.

Based on the use of fuelwood to cure tobacco leaves and the average price of fuelwood sold in the market, the total wood consumption for curing tobacco leaves was converted to carbon emission and consequent cost on the environment given by BDT 26,199.87 (equivalent to USD 310.06) per acre of land used for tobacco cultivation. The steps involved in the estimation are illustrated in Table 12.

### 4.6. Social Benefit (net) of Tobacco Cultivation

As shown in the Table 13, the private financial profit from tobacco cultivation is BDT 46,259 per acre of land used for tobacco cultivation. When the opportunity cost of unpaid family labour, health costs, opportunity cost of owned and sharecropped land and the opportunity cost of using fuel wood from own sources are included in the cost of production, the private economic profit comes down to BDT 45,713 per acre, indicating sheer loss to tobacco farmers. This loss is even greater for contract farmers than for independent farmers. Subtracting the environment costs from the private economic profit, the net social benefit amounts to BDT −77,411 per acre (equivalent to USD −916.11). This implies that the society is incurring a huge loss from tobacco cultivation. A larger net loss on society is inflicted by the contract farmers compared to the independent farmers.

All the cost differences among various groups presented in the previous tables are descriptive. Therefore, these could have been tagged as sampling variations. However, we have also applied statistical tests (t-tests) to verify whether that is the case. As shown in Table 14, it turned out that the differences are, in fact, statistically significant. Even though tobacco farmers have had higher accounting profit compared to non-tobacco farmers which is also significant, this situation got reversed once implicit costs are taken into consideration. However, contract farmers tend to have higher accounting and economic profits in comparison with independent tobacco farmers, and these differences are significant in both cases.

### 4.7. Farmers’ Perceptions about Tobacco Farming

The top three reasons why former tobacco farmers discontinued tobacco cultivation were extensive labour requirements, the availability of alternative livelihood options, and negative effects on land fertility (Table 15). While asked about the reasons for not producing tobacco, the top three reasons were extensive labour requirements, non-suitability of land for tobacco cultivation, and negative effects on land fertility (Table 15).

Both tobacco and non-tobacco farmers believed that tobacco cultivation requires relatively more chemical fertilizers compared to other crops. In addition, tobacco plants attract more pests and insects and therefore larger amounts of chemical pesticides are being used for tobacco cultivation. Despite such knowledge and perceptions, some farmers continue to grow tobacco. Based on the responses of tobacco farmers on what motivate them to continue to grow tobacco, the key drivers appeared to be easy access to market and high profitability that topped the list of the number one influencing factor (Table 16). Thus, it turns out that, despite the higher profitability of alternative crops as observed in this study, easy access to market dominates the decision of tobacco farmers to continue tobacco cultivation. Ensuring higher profit margin in other crops without creating opportunity for easy market access may not be highly effective in inducing tobacco farmers to switch to cultivation of alternative crops. There are other motivating factors including incentives from tobacco companies (e.g., buyback guarantee) that do not necessarily exist in the case of other crops.

## 5. Discussion

With a well-designed survey and analytical approach, the current study fills a significant knowledge gap in measuring the economic costs of tobacco cultivation. The study provides concrete evidence against the most-circulated myth of prosperity offered in tobacco cultivation. Most importantly, the study finds that tobacco cultivation is not a profitable undertaking for farmers if the opportunity costs of unpaid family labour and own land and other resources are taken into consideration. Even though the financial profit from tobacco cultivation is found to be positive, it is much lower than that gained from alternative crops. Furthermore, when the environmental costs posed by tobacco cultivation to water, soil, and forest resources are considered, the broader society incurs a huge net loss from tobacco cultivation.

Tobacco cultivation contributes to the short- and long-term loss of soil quality, water quality, and deforestation, thus slowing down the current income of the farmer as well as increasing the life cycle cost through loss of life cycle income. This in turn affects inter-generational equity and slows down economic progress of the community dependent on tobacco cultivation for their livelihood, in both the short and the long run.

Contract farming is a common feature in tobacco cultivation in Bangladesh, although the prevalence is low compared to Indonesia, Kenya, and Zambia. The low number could be due to the strict eligibility requirement of tobacco companies. Anecdotal evidence suggests that independent farmers can sell tobacco leaf through other contract farmers. Moreover, since contract famers have to employ both extensive and intensive labour hours, some tobacco farmers might be reluctant to get involved in contracts with tobacco companies.

Contract famers are different from the independent farmers in many respects. They tend to spend significantly more time in farming and a significantly higher proportion of family labour is used in contract farming. Though they earn larger financial profit than the independent farmers, if opportunity costs of unpaid family labour and other own resources are taken into account, their profitability gets lower. Nevertheless, they continue contract farming due to the availability of various incentives (e.g., marketing opportunities, advance credit services) from tobacco companies. A similar conclusion is drawn in the studies of tobacco farming in Kenya [7], Indonesia [8], Zambia [9], and Malawi [31].

One major limitation of this study is that the estimates of private and social costs and profitability were obtained per acre of land used for cultivation. In the absence of reliable data on the aggregate acreage of land used for tobacco cultivation, a national level estimate of farmer’s and society’s costs and benefits could not be obtained. However, by making the per acre estimates available, this study creates the scope for making national level estimates in the future when data on total area of tobacco cultivation, such as a proper agricultural census, becomes available. In addition, we did not cover all the diseases attributable to tobacco farming due to limited resources-time and fund-wise. As the disease-specific health cost has not been estimated, the retrospective reporting of treatment cost can potentially be an understatement. Hence, reporting that cost would be an understatement of the true cost. That is one of the limitations of the paper which calls for further research to depict the incidence of tobacco farming related diseases and disease-specific cost to get a comprehensive estimate of the health cost.

## 6. Conclusions

Tobacco cultivation poses a significantly high environmental cost that causes a net loss to society. From a household survey of current, former, and never tobacco farmers, with laboratory tests of soil and water of the respective areas, this study estimated the financial and economic profitability, accounting for implicit costs, as well as environmental costs per acre of land used for tobacco cultivation. The study finds that it is less profitable than other crops for the farmers and turns into a losing concern when the opportunity cost of unpaid family labour and other owned resources and the health effects of tobacco cultivations are included. Nevertheless, the availability of unpaid family labour and the options of advanced credit as well as a buy back guarantee from the tobacco companies attract farmers to engage in and continue tobacco cultivation. Supply side interventions to curb the tobacco epidemic in Bangladesh need to address these factors to correct the wrong incentives and motivate tobacco farmers to switch to alternative livelihood options.

## Figures and Tables

**Figure 1 ijerph-17-09447-f001:**
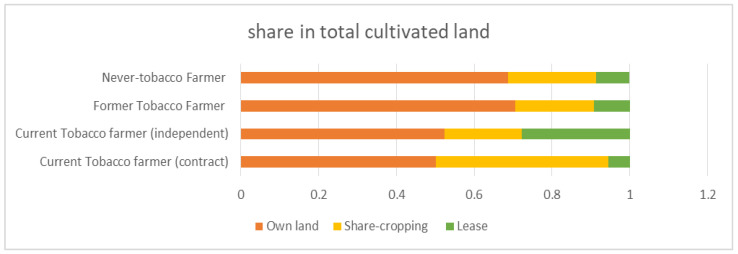
Composition of total cultivable land by ownership status.

**Table 1 ijerph-17-09447-t001:** Sample size of current, former and never tobacco farmers in tobacco-growing and non-tobacco-growing sample areas.

Area	District	Current Tobacco Farmer	Former Tobacco Farmer	Never Tobacco Farmer	Total
Tobacco-growing districts	Manikgonj	148	86	0	234
	Bandorban	166	81	120	367
	Kustia	161	78	115	354
	Lalmonirhat	150	82	122	354
	Sub-total	625	327	357	1309
Non-tobacco-growing districts	Barisal	-	-	60	60
	Sylhet	-	-	60	60
	Mymensingh	-	-	60	60
	Rajshahi	-	-	60	60
	Sub-total	-	-	240	240
Total		625	327	597	1549

**Table 2 ijerph-17-09447-t002:** Characteristics of the farmer households.

Type	Maximum Education	Household Size (Age > 12)	Household Size	Primary Occupation
Tobacco farmer	9.18	3.34	4.70	Farmer (79.24%)
*Contract farmer*	9.40	3.52	5.22	Farmer (75.25%)
*Independent farmer*	9.07	3.26	4.47	Farmer (81.51%)
Former tobacco farmer	9.46	3.35	4.62	Farmer (72.31%)
Never tobacco farmer	8.97	3.35	4.81	Farmer (75.95%)
*Tobacco area*	8.88	3.31	4.79	Farmer (75.11%)
*Non-tobacco area*	9.23	3.46	4.88	Farmer (78.26%)

**Table 3 ijerph-17-09447-t003:** Socioeconomic status of the sample households.

Type of Farmer	Low	Medium	High	N
Tobacco farmer	211	230	205	646
(%)	33	36	32	100
*Contract farmer*	61	66	72	199
(%)	31	33	36	100
*Independent farmer*	147	163	132	442
(%)	33	37	30	100
Former tobacco farmer	96	125	98	319
(%)	30	39	31	100
Never tobacco farmer	235	159	210	604
(%)	39	26	35	100
*Tobacco area*	174	137	136	447
*(%)*	39	31	30	100
*Nontobacco area*	61	22	74	157
*(%)*	39	14	47	100

**Table 4 ijerph-17-09447-t004:** Land holding (decimals) of tobacco and never tobacco farmers.

	Current Tobacco Farmer					
Variable	N	Own Land	Share-Crop	Lease	Govt. Khas Land.	Total
Land size	642	83.90	39.74	22.33	0.00	145.96
*Contract farmer*	199	111.23	76.24	8.93	0.00	196.40
*Independent farmer*	443	71.80	23.45	28.48	0.00	123.73
Land cultivated	642	76.24	39.26	22.49	0.00	138.00
*Contract farmer*	199	99.80	75.34	8.93	0.00	184.07
*Independent farmer*	443	65.82	23.16	28.71	0.00	117.69
Land with tobacco cultivation	642	60.67	35.42	21.69	0.00	117.78
*Contract farmer*	199	80.28	70.85	8.93	0.00	160.07
*Independent farmer*	443	51.95	19.59	27.55	0.00	99.10
Land for other crops	642	20.32	9.23	1.20	0.00	30.75
*Contract farmer*	199	29.36	9.35	0.40	0.00	39.11
*Independent farmer*	443	16.33	9.21	1.57	0.00	27.11
	**Former Tobacco Farmer**					
Land size	319	78.77	19.84	9.31	0.03	107.95
Land cultivated	319	68.98	19.72	8.99	0.03	97.72
Land with tobacco cultivation	319	0.00	0.00	0.00	0.00	0.00
Land for other crops	319	57.81	17.23	6.88	0.03	81.95
	**Never Tobacco Farmer**					
Land size	604	108.47	33.36	12.78	0.20	154.80
Land cultivated	604	100.43	32.99	12.47	0.17	146.06
Land with tobacco cultivation	604	0.00	0.00	0.00	0.00	0.00
Land for other crops	604	98.16	32.57	11.57	0.17	142.46

**Table 5 ijerph-17-09447-t005:** Household income, expenses and asset (BDT) by farmer type.

Type of Farmer	N	Monthly Income	Monthly Expense	Asset	Loan from Company
Tobacco farmer	646	19,700	13,743	159,367	
*Contract farmer*	197	16,972	6598	206,764	19,743
*Independent farmer*	445	21,057	2457	139,078	
Former tobacco farmer	319	19,589	3386	49,230	
Never tobacco farmer	604	20,562	1811	70,703	
*Tobacco area*	447	19,068	12,067	167,487	
*Nontobacco area*	157	24,814	1081	179,860	

**Table 6 ijerph-17-09447-t006:** Average non-farm employment (hours) and monthly income (BDT) of households.

Type of Farmer	Month/Year	Days/Month	Hours/Day	Monthly Pay	Annual Hours
Current tobacco farmer	10.361	21.374	7.082	6453.026	1706.072
*Contract farmer*	9.459	21.426	7.918	10295.08	1689.492
*Independent farmer*	10.511	21.395	6.904	5707.484	1706.81
Former tobacco farmer	11.014	21.722	7.198	7832.051	1884.142
Never tobacco farmer	11.302	20.663	6.436	5602.403	1702.678
*Tobacco area*	11.602	19.741	6.363	5050.154	1613.842
*Nontobacco area*	10.556	23.286	6.663	7209.184	1937.459

**Table 7 ijerph-17-09447-t007:** Average cost of inputs per household in per crop terms for tobacco/alternative crop production by farmer type.

Type	No of Obs.	Fertilizer Cost	Pesticide Cost	Other Variable Input Cost	Durable Input Cost	Misc. Cost	Total Input Cost
Current tobacco farmer	646	14,288	1926	23,523	7312	128	47,176
*Contract farmer*	197	19,218	2912	31,988	10,051	188	64,358
*Independent farmer*	445	12,202	1500	19,932	6147	102	39,884
Former Tobacco Farmer	319	5638	796	7150	5034	67	18,685
Never tobacco farmer	604	6941	1296	13,490	4520	711	26,956
*Tobacco area*	447	6721	1157	13,486	4097	961	26,423
*Nontobacco area*	157	7566	1690	13,502	5723	-	28,481

**Table 8 ijerph-17-09447-t008:** Cost of labour per household by source and farmer type.

Type of farmers	No of Obs.	Family Hour	Hired Hour	Total Person Days	Wage Rate	Total Labour Cost
Current tobacco farmer	646	1656	700	295	462	136,012
*Contract farmer*	197	2046	1231	410	587	240,270
*Independent farmer*	445	1499	471	246	411	101,126
Former tobacco farmer	319	952	235	148	319	47,319
Never tobacco farmer	604	934	288	153	432	65,930
*Tobacco area*	447	971	272	155	413	64,226
*Nontobacco area*	157	829	334	145	483	70,263

**Table 9 ijerph-17-09447-t009:** Cost of production, sales revenue and profit per acre of land by type of farmers.

Type of farmer	(BDT)	(BDT) ^1^	(BDT)	(%)
Tobacco farmer	76,064.21	92,527.56	16,463.35	22%
*Contract farmer*	97,080.98	136,558.56	39,477.58	41%
*Independent farmer*	67,244.89	73,035.19	5790.30	9%
Former tobacco farmer	29,172.09	73,605.92	44,433.84	152%
Never tobacco farmer	29,388.58	63,627.17	34,238.59	117%
*Tobacco area*	27,394.20	77,533.16	50,138.96	183%
*Nontobacco area*	35,875.46	28,999.05	−6876.41	−19%

Note: Cost includes input cost, hired labour cost and land rent cost as a share of per acre of land used for tobacco cultivation. ^1^ This cost includes opportunity cost of unpaid family labor.

**Table 10 ijerph-17-09447-t010:** Effects on health and health costs.

Type of Farmer	Sick Days	Lost Days	Lost Income	Doctor’s Fee	Medicine Cost	Hospital Cost	Transport Cost to Hospital	Other Cost	Total Direct and Indirect Cost
Current tobacco farmer	5.32	3.08	1380.40	100.21	517.07	249.57	57.70	33.46	2338.42
*Contract farmer*	3.46	2.06	920.32	111.17	509.15	84.26	65.03	37.95	1727.88
*Independent farmer*	6.15	3.54	1583.41	96.03	522.70	325.00	54.98	31.78	2613.90
Former tobacco farmer	7.58	4.52	1393.81	170.89	749.02	648.53	125.69	63.64	3151.57
Never tobacco farmer	4.07	2.64	1091.70	91.23	424.78	395.38	51.13	51.24	2105.45
*Tobacco area*	4.43	2.79	1155.86	88.08	436.33	389.28	46.38	26.29	2142.22
*Nontobacco area*	3.04	2.20	909.02	100.17	391.90	412.74	64.65	122.29	2000.77

**Table 11 ijerph-17-09447-t011:** Cost of processing/curing of tobacco leaf.

Type of farmer	Tobacco Leave Curing Process		Types of Fuel Used to Cure Tobacco Leaves		Sources of Fuel			Wood Required/40 kg of Leaves (kg)	Avg. Price Per 40 kg Wood
	Sun Dry	Fuel Dry	Wood	BAMBOO	Own Source	Market	Landlord		
All tobacco farmers	52.17%	47.83%	94.24%	1.44%	7.19%	76.98%	11.51%	201.78	147.45
Contract tobacco farmers	36.02%	63.98%	96.72%	0.82%	13.93%	66.39%	14.75%	203.13	124.34
Independent tobacco farmers	59.95%	40.05%	91.95%	2.01%	2.01%	86.58%	8.05%	201.15	165.57

**Table 12 ijerph-17-09447-t012:** Environmental cost of tobacco curing per acre of land.

Item	Current Tobacco Farmers	Contract Tobacco Farmers	Independent Tobacco Farmers
**Average production (mt) per household**	1.29	1.88	1.03
**Average cultivated land size (decimal) per household**	117.78	160.07	99.10
**Average production per acre (MT)**	1.10	1.17	1.04
**Wood required (MT) to cure per MT of leaves**	5.04	5.08	5.03
**Wood required to cure tobacco leaves per acre (MT)**	5.52	5.96	5.23
**Woods used to cure tobacco (MT) per acre (Only 47.83%, 63.98% & 40.05% are fuel cured)**	2.52	3.48	2.07
**Carbon emission (MT) per acre of tobacco cultivation from burning: 50% × 4.2929**	1.26	1.74	1.03
**Carbon emission from deforestation**	1.09	1.50	0.89
**Total Carbon emission**	2.35	3.24	1.92
**Total CO2 emission**	8.61	11.87	7.06
**@ $36 per ton of CO2 (USD)** [1]			
**Total environmental cost per acre (USD)**	310.06	427.43	254.33
**Total environmental costs per acre (BDT)**	26,199.87	36,117.77	21,490.67

Note: Since some of the social costs will be incurred in future, a discount rate is required to estimate the present value of the future costs. A larger discount means more weight is given on the present than the future costs. In literature 3% and 5% discount rate are used frequently. We used 5% discount rate to make sure that our results do not over-estimate the social costs. Therefore, the current study provides a conservative estimate.

**Table 13 ijerph-17-09447-t013:** Annual social net benefit from tobacco cultivation per acre of land use.

Item	All Tobacco Farmers	Contract Tobacco Farmers	Independent Tobacco Farmers
**Sales revenue (R)**	122,322.82	150,136.22	108,572.52
**Direct input costs (C)**			
*Material input cost*	40,054.68	40,206.16	40,246.45
*Hired labour costs*	34,302.44	56,376.07	24,408.44
*Land rental cost*	1707.10	498.75	2590.00
***Private financial profit (P = R-C)***	46,258.61	53,055.24	41,327.63
**Indirect costs (D)**			
*Opportunity cost of family labour*	81,176.92	93,727.07	77,635.64
*Opportunity cost of land (owned and shared)*	8822.50	10,030.85	7939.60
*Opportunity cost of fuelwood from own sources*	280.58	417.35	156.10
*Income loss due to sick days*	1691.92	938.49	2218.08
***Private economic profit (F = P-D)***	−45,713.32	−52,058.53	−46,621.79
*Cost of firewood for tobacco leaf curing (E1)*	26,199.87	36,117.77	21,490.67
*Cost of treating soil and water contamination(E2)*	5498.00	5498.00	5498.00
**Net social benefit (B = F-E1-EG2), in BDT**	−77,411.19	−93,674.30	−73,610.46
Net social benefit, in USD ^1^	−916.11	−1108.57	−871.13

^1^ Using the exchange rate of 1 USD = 84.5 BDT (Bangladesh Bank, 5 September 2019).

**Table 14 ijerph-17-09447-t014:** Profitability of Tobacco farming compared to alternatives across farmer types.

Type of Farmer	Mean Difference (Per 1/3 of Acre)	N	df	*t*-Statistics
**Accounting Profit (Nominal cost-benefit)**				
Tobacco farmer vs non-tobacco farmer	12,141.07	1238	1236	14.46 ***
Tobacco farmer vs former tobacco farmer	11,830.68	954	952	10.96 ***
Former tobacco farmer vs non-tobacco farmer	310.3898	886	884	0.33
Contract farmer vs independent farmer	13,515.5	635	633	10.58 ***
**Economic Profit (Social cost-benefit)**				
Profitability of Tobacco farming compared to alternatives across farmer types	−15,513.93	1232	1230	−27.71 ***
Tobacco farmer vs former tobacco farmer	−22,843.21	948	946	−20.38 ***
Former tobacco farmer vs non-tobacco farmer	310.3898	886	884	0.33
Contract farmer vs independent farmer	13,515.5	635	633	10.58 ***

Note: *** means significant at 1% level of significance

**Table 15 ijerph-17-09447-t015:** Reasons for discontinuing tobacco production by former tobacco farmers and not producing tobacco by never tobacco farmers.

Former Tobacco Farmers	%	Never Tobacco Farmers	%
Top three reasons for discontinuation		Top three reasons for not cultivating tobacco	
Extensive labour requirement	26%	Extensive labour requirement	34%
Availability of alternative livelihood options	20%	Non-suitability of land for tobacco cultivation	32%
Negative effects on land fertility	18%	Negative effects on land fertility	23%

**Table 16 ijerph-17-09447-t016:** Influencing factors for tobacco cultivation.

Factors	Preferred Factors		% Claimed as 1st Reason	% Claimed as 2nd Reason	% Claimed as 3rd Reason
Attractive features	N	(%)	(%)	(%)	(%)
Easy access to market	593	96.4	29.2	20.3	21.3
Only way to survive	477	77.6	15.3	17.6	7.7
Feel comfortable with this	508	82.6	6.8	18.3	19.2
Easy availability of land	462	75.1	5.7	8.7	8.2
Encouraged by other farmers	512	83.3	4.7	9.4	15.3
Good incentive from the tobacco company	431	70.1	5.2	5.0	5.8
Profitable venture	541	88.0	23.3	13.3	13.7
Way to repay loan received from the tobacco company	292	47.5	6.0	6.1	6.7
Option of receiving timely training	161	26.2	0.3	0.8	1.5
Other reasons	45	7.3	3.5	0.3	0.7
	600		100	100	100

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
