# Peer review of "The Economic Cost of Tobacco Farming in Bangladesh"

_ijerph, 2020, doi:10.3390/ijerph17249447_

Round 1

Reviewer 1 Report

Due to lack of scientific evidence considering the harmful effects of tobacco cultivation on the environment, this study contributes to the body of knowledge by measuring the environmental costs as well as other implicit costs such as unpaid family labor and health costs of tobacco cultivation.  The paper is clearly presented with a strong motivation statement.  The methodology used to calculate the environmental cost of tobacco farming is scientifically solid.  I only have a couple of minor comments as the following.

1. It seems that there are health hazards for tobacco growers, which makes the consideration of health costs of tobacco cultivation meaningful in the present study. What the health hazards for growers of tobacco are, however, were not depicted in this work.  Please add the discussion to the introduction.

2. The analytical framework is well organized which gives a detailed description of the measures of costs. However, the section “3.3. Household level profitability of tobacco farming” is relatively short and did not provide enough information on how the implicit costs including opportunity costs of unpaid family labour and own land and fuelwood from own sources are measured.  Moreover, in the same section, implicit cost such as unpaid family labor and health costs of tobacco cultivation stressed in the abstract and introduction were not described.

3. The analysis in the paper is basically descriptive. In order to identify the association of tobacco cultivation and its effects on the environment, grower’s health, etc., it is strongly suggested to run statistical tests of differences in all sorts of costs and financial profits between the three groups of grower.

4. Current tobacco farmers are divided into two groups, contract and independent, in the analysis. Is contract farming regarded as a common feature of the tobacco farming sector under study? Any specific reason to characterize tobacco growers by contract farming?  I would also suggest to the authors to run statistical tests of differences in all sorts of costs and financial profits between the contract and independent growers.

5.The conclusion section should give a summary of the results to support statement such as:

  • Tobacco cultivation is less profitable than other crops for the farmers.
  • Tobacco cultivation turns into a losing concern when the opportunity cost of unpaid family labour and other owned resources and the health effects of tobacco cultivations are included.

6. Typo in the first paragraph of “3.3. Household level profitability of tobacco farming”:

“The private eeconomic profit from tobacco farming was estimated ….”

Author Response

Dear Reviewer,

I would like to express my gratitude for your comments which were useful for further development. We have thoroughly gone through your insightful comments and made recommended changes.

Please find the edited version with the track-change attached. Please also find our responses to your comments in the attachment.

Kind regards,

AKM Ghulam Hussain

Reviewer 2 Report

The extent of tobacco cultivation remains substantially high in Bangladesh which is the 12th largest tobacco producer in the world. Using data from a household survey of current, former and never tobacco farmers based on a multi-stage stratified sampling design with a mix of purposive and random sampling of households, this study estimated the financial and economic profitability per acre of land used for tobacco cultivation. I think the paper is reasonably well written and I found it very interesting to read. However, I would like to give some minor comments as follows:

Authors should consider the maximum value on the horizontal axis of figure 1 to be 1 instead of extending the value to 1.2

Table 10. The width of this table should be increased in order to accommodate the column labels

Table 11. Tobacco leaf (not leave) curing process

Table 15 is not explicit enough. How did the authors categorise the influencing factors for tobacco cultivation as No1 reason, No2 reason and No3 reason? This is not easy to comprehend. Authors might need to relabel it for increased clarity.

Author Response

(The authors gave the same response as above.)
